# Soft Robots’ Dynamic Posture Perception Using Kirigami-Inspired Flexible Sensors with Porous Structures and Long Short-Term Memory (LSTM) Neural Networks

**DOI:** 10.3390/s22207705

**Published:** 2022-10-11

**Authors:** Jing Shu, Junming Wang, Sanders Cheuk Yin Lau, Yujie Su, Kelvin Ho Lam Heung, Xiangqian Shi, Zheng Li, Raymond Kai-yu Tong

**Affiliations:** 1Department of Biomedical Engineering, The Chinese University of Hong Kong, Hong Kong SAR 999077, China; 2Department of Aeronautics, Imperial College London, London SW7 2BX, UK; 3Department of Rehabilitation Sciences, The Hong Kong Polytechnic University, Hong Kong SAR 999077, China; 4Department of Surgery, The Chinese University of Hong Kong, Hong Kong SAR 999077, China

**Keywords:** soft robotics, flexible sensors, kirigami-inspired structures, flexible porous structures, deep learning, long short-term memory neural network

## Abstract

Soft robots can create complicated structures and functions for rehabilitation. The posture perception of soft actuators is critical for performing closed-loop control for a precise location. It is essential to have a sensor with both soft and flexible characteristics that does not affect the movement of a soft actuator. This paper presents a novel end-to-end posture perception method that employs flexible sensors with kirigami-inspired structures and long short-term memory (LSTM) neural networks. The sensors were developed with conductive sponge materials. With one-step calibration from the sensor output, the posture of the soft actuator could be calculated by the LSTM network. The method was validated by attaching the developed sensors to a soft fiber-reinforced bending actuator. The results showed the accuracy of posture prediction of sponge sensors with three kirigami-inspired structures ranged from 0.91 to 0.97 in terms of $R^{2}$. The sponge sensors only generated a resistive torque value of 0.96 mNm at the maximum bending position when attached to a soft actuator, which would minimize the effect on actuator movement. The kirigami-inspired flexible sponge sensor could in future enhance soft robotic development.

## 1. Introduction

In recent years, with the development of compliant materials, soft robotics has been a research hotspot, especially for robots inspired by nature [1]. Soft robots have significant advantages in human–robot interactions thanks to their naturally deformable structures. Soft sensors were also developed to detect the deformation of soft robots (including curvature [2], tensile or shear [3]) and achieve better control performance. With the use of low-tensile modulus materials, the embedded soft sensors have a minimal effect on impedance changes to surrounding structures [1]. By co-considering the soft sensor information and characteristics of the soft actuators, the posture and generated force/torque could be estimated simultaneously [4].

There are two main categories of on-board soft sensors based on the working mechanisms: (1) soft sensors that make use of electrical properties (piezoelectric effects [5], piezoresistive effects [6], capacitance [7]); (2) soft sensors make use of optical properties [8,9]. In this paper, we mainly focus on the sensors that make use of piezoresistive effects.

The mechanical properties of soft on-board/embedded sensors matter, since their sensing capability depends on the strain limits. Researchers improve the measuring ranges of soft sensors by using highly stretchable materials during sensor fabrication [10], employing buffering-by-buckling structures [11] during flexible sensor design, or combining both organically. In [12], researchers mentioned several typical buffering-by-buckling structures that are beneficial to soft sensor design: (1) In-plane or coiled stretchable configurations: sensors were bent [13,14,15] or twisted in 3D [16,17] to increase the overall deformation ratio (significantly larger than the fracture strain of sensing material). (2) Kirigami abd open-mesh structures [18,19,20,21,22,23,24,25,26]: these two structures are similar. The patterns for cutting were designed to be applied to lamellate materials. During the deformation, the opening/cutting edge would deform to accommodate the overall sensor deformation; (3) 3D porous structure [27,28,29,30]: similarly to the mechanisms of kirigami/open-mesh structure, cavities inside the structure would be compliant with the overall sensor deformation, which increases the stretching capabilities of the sensors. The strategies mentioned above could be combined further to improve the stretching properties of the designed sensors (e.g., kirigami-inspired structure and highly stretchable materials were combined in [22,23]). In this paper, we combined the kirigami-inspired structure and the 3D porous structure to change the resistance of an off-the-shelf sponge material, with respect to the tensile deformation of the sensor. Kirigami-inspired structures were employed to strengthen the stretching performance. Thanks to the porous structure of the sponge material, the tensile resistance values of the sensors were controlled at a low level when compared with sensors made from other materials. Therefore, the influence of the sensor on the attached soft actuators’ motion is negligible.

Sensor calibration is an important issue during the sensor design process. In this work, a novel long short-term memory (LSTM) neural network was used to perceive the strain of the attached objects. LSTM neural networks are unique recurrent neural networks (RNN) that can deal with sequential data, and such methods have already been adopted in soft sensor calibration for pressure/force sensing [31], robot posture and control [4,22,32,33] and tactile localization [34]. This neural network can calibrate the sensor in one step when there is a reliable training dataset [4] while solving the problem with the kirigami-inspired sensors of nonlinearity, hysteresis and stress relaxation properties [35]. By using the sponge material as the sensing material and training the calibration neural network on sensing signals from two kirigami-inspired flexible sensors distributed on the actuator’s side wall, the bending angles of the actuator could be predicted.

This paper describes a set of flexible strain sensors incorporating a kirigami-inspired structure that were made from a piece of conductive sponge material, and a machine learning approach was used for calibration. In Section 2.1, the design and fabrication of the sensors and actuators are illustrated, including the kinematic description of the sensor and the structure of the calibration neural network. The experimental setup and results are shown in Section 2.2 and Section 3, respectively. At the end of the paper, the limitations and potential applications of the sensor are discussed.

## 2. Materials and Methods

### 2.1. Design Overview and Justification

#### 2.1.1. Design and Fabrication of Sensors with Kirigami-Inspired Structures and Soft Fiber-Reinforced Bending Actuators (FRBAs)

The kirigami-inspired structure and 3D porous structure were combined in our design to increase the stretching ability and reduce the tensile resistance. Classical flat-cutting kirigami patterns were designed using AutoCAD (Autodesk, Inc., San Rafael, CA, USA) with the main body dimensions of 15 × 20 mm. Two connector squares on the two ends of the main body were designed for the wire connection (Figure 1). By changing the distance between cutting edges, three kirigami patterns (structures with two units, three units and four units) were designed. A layer of 1 mm-thick off-the-shelf conductive sponge (Beilong electronics, Guangzhou, China) formed the main body for each sensor. During the deformation of the sensor, the resistance of the conductive sponge changes. By measuring this resistance change, the deformation of the attached object is captured. Under the conductive sponge layer, there is a 25 μm-thick polyimide film (Kapton, Dupont, Wilmington, DE, USA), which has excellent mechanical properties and is used to sustain the external stretching and elongate the service life of the sensor. The bottom side of the polyimide film sticks to the attached soft actuator. Since the polyimide film is almost unstretchable, during the stretching of the sensor, the conductive sponge layer is bent along the attached polyimide film instead of being directly stretched along the elongation direction of the sensor. Therefore, the main sponge body is not directly in contact with the attached actuators.

The sensors were fabricated based on the planar fabrication paradigm [36,37,38]. Laminates of materials and fabrication sequences are shown in Figure 2. A ${\mathrm{CO}}_{2}$ laser cutter (CMA960, YUEMING Laser, China) was employed to cut the desired patterns on the laminate materials to ensure the consistency of the sensor’s performance. Location holes and well-fitted steel location pins were employed to improve the assembly accuracy further. A 170 μm, double-sided, pressure-sensitive adhesive (300LSE, 3M Company, Maplewood, MN, USA) layer was used to integrate the polyimide film and conductive sponge layer. The same adhesives were employed to stick the sensors to the attached objects. After the assembly, brass electrodes were stick to the two ends of sensor bodies using silver conductive adhesives (3703, Sinwei, China); the servo structure (shown at the top of Figure 1) of the sensors was then removed; and the sensor’s body was stuck to actuators using silicon adhesives (Sil-poxy™, Smooth-On, Inc., Macungie, PA, USA). The photos of actuators with kirigami-inspired flexible sensors are shown in Figure 3.

A soft fiber-reinforced bending actuator was designed and fabricated to test the performance of the kirigami-inspired sponge bending sensor. The design and fabrication of the actuator were the same as in our previous work [39] The actuator body was made from silicon elastomer (Dragon Skin 30™, Smooth-On, Inc., USA). Flexible reinforcement fibers (Figure 3) were wound along the rectangular actuator body with a double helix winding pattern. A thin steel sheet was placed on the bottom surface of the actuator to act as an unstretchable layer. When the soft actuator was pressurized, serried reinforcement fibers along the actuator length would prevent the actuator’s expansion and provide a trend to extend along the actuator’s length. Since the unstretchable steel sheet provides restraint, the side of the actuator without the constraint would be extended, and the actuator would finally bend to the side with the constraint.

#### 2.1.2. Kinematic Description of Sensors Integrated with Bending Actuators

The relationship between the deformation of the kirigami-inspired flexible sensors and the bending angles of the actuator was developed. A schematic diagram of the actuator integrated with sensors is shown in Figure 4. Since a soft fiber-reinforced bending actuator is commonly used in rehabilitation applications [39,40], a schematic finger with joints is also presented. The bending actuator could be simplified using either the constant curvature (CC) model or piecewise constant curvature (PCC) model. When the actuator is with the PCC model, it would be divided into two pieces, where the boundary is the contacting point of the finger’s proximal interphalangeal (PIP) joint. Therefore, for every bending angle $\theta_{i}$ (i=1,2), we have that
(1)$$r_{i}=L_{i}/\theta_{i},$$
where $r_{i}$ is the radius of the actuator arc for each piece. $L_{i}$ is the length of the actuator piece’s bottom side. Thanks to the employment of stretchable layer, $L_{i}$ is a constant value.

Since there is an unstretchable steel layer, the length of the actuator’s bottom side would be constant during deformation. When the thickness of the actuator is *h*, for the length of the actuator’s top side $L_{Top_{i}}$, we have
(2)$$L_{Top_{i}}=\left(r_{i}+h\right)\cdot \theta_{i}.$$
Since the kirigami-inspired flexible sensors are firmly attached to the side wall of the actuator, the deformation ratio of the actuator’s top side would be identical to the deformation ratio of the sensor. The sensor’s deformation ratio $\epsilon_{i}$ (i=1,2) was defined as $\epsilon_{i}=\left(L_{Top_{i}}-L_{i}\right)/L_{i}=\left(D-D_{0}/D\right)$. When the actuator is using a constant curvature model, i=1 only. $\theta_{1}$ and $\theta_{2}$ in Figure 4 would be combined together and form the overall bending angle θ of the actuator. $L_{1}$ and $L_{2}$ would be combined to have *L* represent the overall length of the unstretchable layer.

In our design, we have *L* = 75 mm and *h* = 16 mm. Based on Equations (Equation 1) and (Equation 2), the sensor deformation ratio ϵ would be 33.5% when the overall bending angle $\theta =90^{\circ }$.

#### 2.1.3. Design of a Neural Network for Sensor Calibration

In this section, the design of the calibration neural network is presented. In Section 1, we mentioned it was necessary to use a data-driven method to calibrate the sensor made of a hyperelastic material that could handle the problem of hysteresis. In this work, a LSTM calibration neural network was designed to map the resistance of sensors to the bending angle of the soft bending actuator. The structure of the neural network is shown in Figure 5. It comprises an input layer, an output layer, and three hidden layers (each contains 100 hidden units). Early stopping (200 epochs) and dropout layers (0.1 dropout rate) are also utilized in the neural network to prevent the problem of overfitting (similar strategies were adopted in [4,22]). The inputs of the neural network are the sensing signals from the sensors. When the soft bending actuator is simplified using the constant curvature (CC) model (i.e., the actuator was treated as a whole part), the output of the neural network would be the overall bending angle of the actuator, and the neural network inputs would be the sensing signals of two sensors (all-to-one mode). If the actuator-simplifying model is the PCC model, there would be two individual LSTM calibration neural networks. For each of them, the inputs are the sensing signals of a specific sensor, and the outputs are the bending angle of corresponding segments (one-to-one mode). In Section 3, the performances of the calibration neural networks in these two modes are illustrated.

### 2.2. Experimental Setup

#### 2.2.1. Determining the Characteristics of Sensors

Experimental Setup to Determine the Mechanical Properties of Sensors

In this section, the experimental setup for determining sensor characteristics is illustrated. It was mentioned in Section 1 that by employing the buffering-by-buckling kirigami structure, the internal stress of the material during stretching would be significantly reduced. To prove this, a benchtop testing platform was built, as shown in Figure 6. The relationship between the sensor’s resisting force and its deformation ratio is determined. The two ends of the sensors were mounted on a customized universal tensile testing machine. One end of the sensor was fixed, and another was connected to a linear stage, where a load cell (SBT674-1KG, Simbatouch, Guangzhou, China) connected the moving stage and the jig. A rope encoder (MPS-XS-1000mm, Miran, Shenzhen, China) was used to measure the displacement. The sensing information from the load cell and the rope encoder was recorded by a DAQ device (USB6212, National Instrument, USA). During the experiment, the sensors with three different kirigami-inspired structures were stretched at 50 mm/min until the deformation reached 8 mm (40% deformation ratio). To embody the advantage of 3D porous structures, a 1mm-thick conductive silicon (KE-3601SB-U, Shin-Etsu Silicon, Tokyo, Japan) sheet used in our previous design [23] was also prepared with the same kirigami pattern and stretched under the same conditions.

Finite Element Analysis (FEA) Setup

Finite element analysis (FEA) was performed to determine sensors’ stress/strain distribution during stretching. The finite element software Abaqus 2021 (Dassault Systemes Simulia Corp, Providence, RI, USA) was used to examine the strain within the kirigami structure. The Yeoh 3rd-order hyperelastic model with the coefficients from Szurgott’s work in 2019 [41] were used in the model of the polyurethane (PU) sponge (presented in Table 1). The polyimide film was modeled as an elastic material with material properties (E = 3.13 GPa, Poisson ratio = 0.3) determined in Qu’s work in 2017 [42]. A nonlinear static simulation was conducted. Consistently with the experiment setup, one end of each of the sensors was fully fixed with an encastre boundary condition, and the other end was stretched with a displacement of 10 mm (50% deformation ratio) in y direction (areas for electrodes were excluded). The positions of the constraints and loads are shown in Figure 7. A tie constraint was set between the sponge layer and the polymide film. The model was simulated with reduced integration (mesh type: C3D8R) and 0.5 mm mesh size.

Experimental Setup to Determine the Resistance Responses of Sensors

The resistance responses of kirigami-inspired flexible bending sensors with respect to the bending angles of the soft actuator were explored. For sensors with the same kirigami pattern, two sensors with identical appearances were stuck to the top and bottom surfaces of the bending actuator (shown in Figure 3) and connected to a voltage divider circuit ($R_{0}$ = 20 Ω). During the experiment, the soft bending actuator was pressurized to 200 kPa and released at three different charging speeds (10, 50 and 100 kPa/s). An ADC module recorded the sensor signals with 16-bit resolution (ADS1115, Adafruit Industries, New York City, NY, USA). The sampling frequency is 20 Hz.

#### 2.2.2. Benchtop Test to Determine Sensors’ Performances and Calibration Neural Network Training

In this experiment, the performances of kirigami-inspired flexible sensors were determined. The experimental setup is shown in Figure 3. The bending angles of the soft bending actuator were recorded by two IMUs (BNO055, Adafruit Industries, New York City, NY, USA) located on the tip and middle of the actuator (recording the proximal interphalangeal (PIP) joint angle and metacarpophalangeal (MCP) joint angle, respectively). The wooden finger model was removed when testing the sensor’s performance and installed during the experiment to predict the finger bending angles. In the experiments, to test the sensors’ performances, three sets of sensors with kirigami patterns mentioned in Section 2.1 were installed and tested. The kirigami sensor with the best performance was selected and used to predict the wooden finger model’s MCP and PIP joint angles. The sensor signal was collected using the methods in Section 2.2.1. The LSTM calibration neural network’s training was performed in the MatlabMachineLearningToolbox. The soft bending actuator was inflated with various pressures for 15 min to get the neural network training dataset (between 0 and 200 kPa; 10 kPa interval), and a variable speed of 10 to 50 kPa/s (10 kPa/s intervals). Then, 70% of the data was used for neural network training, and another 30% was used for neural network testing.

## 3. Results

### 3.1. Characteristics of Kirigami-Inspired Sensors

Mechanical Properties of Kirigami-Inspired Flexible Sensors

The resisting force generated by kirigami-inspired flexible sensors with respect to the deformation ratio is shown in Figure 8. Sensors with two sensing materials and three kirigami patterns were tested. The photos of the stretched sensor bodies are shown in the top left corner of Figure 6. When the distance between the cutting edges becomes narrower (i.e., from two-unit structure to four-unit structure), the resisting force drops significantly. For the kirigami-inspired sensor made from the conductive sponge material, the resistive force rapidly increases when it is close to its elongation limits (two-unit structure in Figure 8a). Although the elastic moduli of PU and conductive silicon are different, the sensors with a 3D porous structure could still reduce the resisting force during stretching, which could cause fewer adverse effects on the motion of the attached actuators.

Finite Element Analysis (FEA) Results

The results of the finite element analysis (FEA) are presented in Figure 9. The strain values are presented in the top left corner of each subfigure. When the cutting edge gap becomes smaller, the strain of conductive sponge material drops, which means the elongation limit negatively correlates with the cutting edge distance. The maximum strains of two, three and four-unit structures were 0.65, 0.33 and 0.25, respectively.

Resistance Response of Kirigami-Inspired Flexible Sensors

The resistance response of the kirigami-inspired flexible sensors is shown in Figure 10. The resistance responses under 10, 50 and 100 kPa/s pressurizing rates are presented using blue, red and yellow curves, respectively. The solid lines show the resistance of the sensors during bending, and the dashed lines present the resistance during actuator relaxation. When the bending angle increased, the resistance of the kirigami-inspired sensors dropped. There was hysteresis in the sensing signals. When the pressurizing rate increased, its resistance value had no significant changes.

### 3.2. Performance of the Calibration Neural Network

As mentioned in Section 2.1.3, the LSTM calibration neural network has two working modes (i.e., all-to-one mode and one-to-one mode). The performance of the calibration neural network in all-to-one working mode is shown in Table 2. The soft actuator bending prediction in a selected time region of three minutes is shown in Figure 11. The kirigami-inspired flexible bending sensor with the three-unit structure achieved the lowest root-mean-square error (RMSE) value (3.85${}^{\circ }$) and the highest coefficient of determination ($R^{2}$) (0.97). When the LSTM calibration neural network is working in the one-to-one mode, the PIP joint angle and MCP joint angle are predicted by flexible sensors installed on the specific segments, separately. The performance of the calibration neural network in this operating mode is shown in Table 3. The finger-joint-angle prediction for the selected time period is shown in Figure 12. The time periods in Figure 11 and Figure 12 are identical for sensors with the same kirigami cutting patterns. It is noted that when two kirigami-inspired sensors located at the top and bottom of the bending actuator work together, the performance of the calibration neural network could be strengthened when compared with the performance of the sensor working individually.

### 3.3. Finger-Joint-Angle Prediction

By comparing the performances of sensors with three kirigami-inspired structures, it could be concluded that the sensor with the “three-unit” kirigami cutting pattern had the best performance. Therefore, the sensors with such kirigami patterns were selected in the experiment for the prediction of the finger-joint angle. The wooden finger model was installed and connected to the soft bending actuator using Velcro. Figure 13 shows a posture photo of the wooden finger model and its predicted posture plotted using the PCC model. The PIP and MCP joint angles are indicated the Figure 13a. The red dashed line is placed vertically up, indicating the positions of the sensors when the actuator is at rest. The IMU placed on the middle and top of the actuator measure the tilting angles of the blue and green dashed lines. The time periods are shown with predicted joint angles for 8, 33 and 79 s. Predicted joint angles with respect to time are shown in Figure 13d.

## 4. Discussion

### 4.1. Resisting Effects of Kirigami-Inspired Sponge Sensors on the Attached Bending Actuator

In our design, the kirigami-inspired sponge sensors were arranged on the tops of soft-bending actuators. For the resistive torque, in our case, we have
(3)τ=F·h,
where τ is the resistive torque generated by the sensor deformation, and *F* is the resistive force (indicated in Figure 8). *h* is the actuator thickness indicated in Figure 4. With this equation, the resistive torques generated by the kirigami-inspired sponge sensors with different kirigami patterns and sensing materials can be calculated, as shown in Table 4. For the resistive torque generated by a kirigami-inspired sponge sensor made from conductive sponge material, there was a 45.26% to 76.56% resistive torque reduction compared with the resistance torque generated by sensors made from conductive silicon material [23].

### 4.2. Selection of the Calibration Tool and Performance of the Kirigami-Inspired Sponge Sensor

According sponge sensor characteristics’ experimental data in Section 3.1, there are hysteresis and nonlinearity in the sensing signals, which are related to the deformation speed of the attached actuators. The LSTM neural network was able to learn those nonlinearity and hysteresis characteristics; it is suitable for our application and was selected as the calibration tool of our sensing system.

The kirigami-inspired sponge sensor could work individually or together by selecting the input information of the LSTM calibration neural network. When more sponge sensors work together, the prediction accuracy will be higher. Based on the performances of sponge sensors with different kirigami patterns, we recommend using the sensors with the pattern “three-unit structure” in a soft rehabilitation glove application, since the sensors with such kirigami patterns had the best performance among the proposed kirigami patterns in our testing results.

### 4.3. Advantages and Limitations of the Kirigami-Inspired Flexible Sensors

There are two main existing challenges/requirements in developing the flexible sensors for soft robot posture perception: (1) the flexible sensor should be flexible enough to minimize any resistive torque that affects the movement of the attached actuators; (2) the sensor should provide high enough accuracy to capture both the dynamic motion and the static posture of the attached actuators.

The main advantage of the technology introduced in this paper is that the combination of a kirigami-inspired structure and 3D porous structure makes the sponge sensor have low resistance to actuator motion and provides excellent performance in stretching. Therefore, the posture prediction accuracy is higher than that of the kirigami-inspired conductive silicon sensor [22,23] and embedded conductive PDMS sensor [4]. The application of planar fabrication methods reduced the fabrication complexity compared to other customized sponge sensors [28,30]. With the spring-like properties of the kirigami-inspired polyimide-based layer and the low viscoelastic properties of porous conductive sponges, the sensor improves the ability to dynamically capture the actuator’s motion when compared with conductive silicon sensors [22].

The limitation of this method is that the sensor is required to be trained with a neural network before application. Data collection from the sensor to cover the measuring space in our current setup was 15 min, and the neural network training was 10 min (PC specification: Intel Core 10700 (CPU), Nvidia GeForce RTX 2080 Super (GPU)). Then, the system was ready to use.

### 4.4. Future Development of the Kirigami-Inspired Sponge Sensor

The sensor’s performance on actuators that can undergo three-degree-of-freedom (DOF) motion can be explored in a future study. Different materials could be compared to enhance the performance of the sensor.

## 5. Conclusions

We used conductive sponge materials with kirigami-inspired structures and LSTM neural networks to monitor the dynamic motion of a bending actuator. Off-the-shelf materials and planar fabrication methods make it more convenient for these techniques to be applied to soft robotic systems. The usage of the calibration neural network provides an end-to-end calibration method and handles the nonlinear and hysteresis problems of the viscoelastic material with acceptable prediction accuracy. Postures for a wooden finger model were predicted, indicating an enormous opportunity to apply the kirigami-inspired flexible sponge sensors to various flexible exoskeleton systems. Overall, the sensory system developed in this paper is significant for soft robot posture perception and closed-loop control. We genuinely expect our work could be adopted and explored by other researchers to benefit the development of soft robotics.

## Figures and Tables

**Figure 1 sensors-22-07705-f001:**
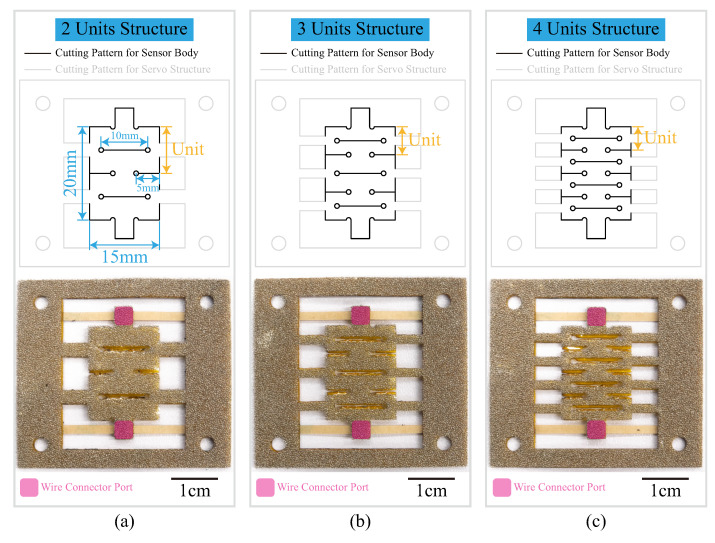
Kirigami-inspired cutting patterns (**top**) and photos of the wire connector ports (**bottom**) of sensors. (**a**) Sensors with two-unit kirigami structure; (**b**) Sensors with three-unit kirigami structure; (**c**) Sensors with four-unit kirigami structure.

**Figure 2 sensors-22-07705-f002:**
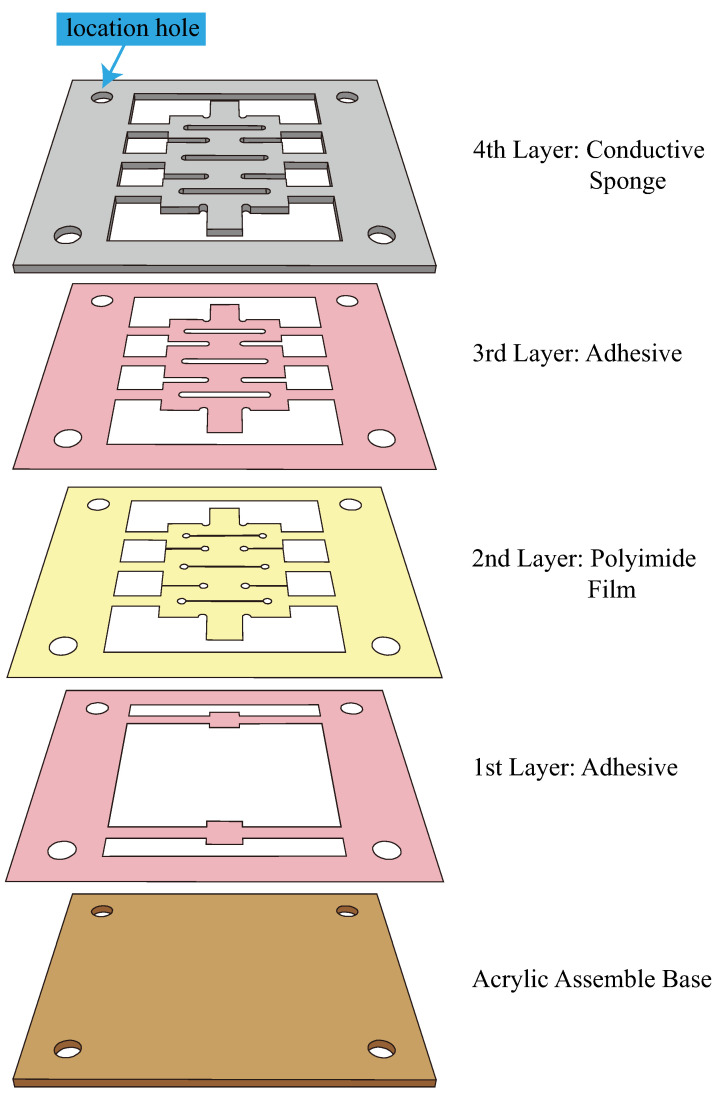
Planer fabrication process of kirigami-inspired flexible sponge sensors. Order of layers and location holes for fabrication are presented.

**Figure 3 sensors-22-07705-f003:**
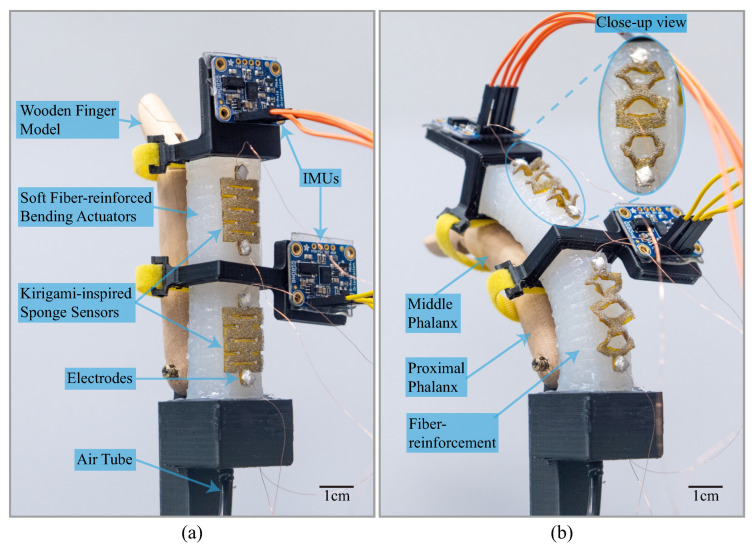
Experimental setup of bending angle perception of a soft pneumatic fiber-reinforced bending actuator by kirigami-inspired flexible sponge sensors. (**a**) The actuator is at resting state; (**b**) the actuator is at pressurized state. A scale bar of 1cm ispresented.

**Figure 4 sensors-22-07705-f004:**
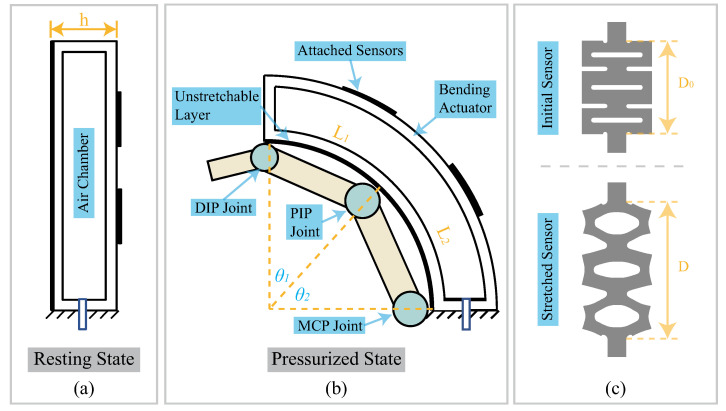
Kinematic description of the bending actuator and kirigami-inspired flexible sensors. (**a**) Bending actuator at resting state; (**b**) bending actuator in a pressurized state with a schematic diagram of the finger model; (**c**) schematic diagram of the initial sensor pattern and the stretched sensor pattern.

**Figure 5 sensors-22-07705-f005:**
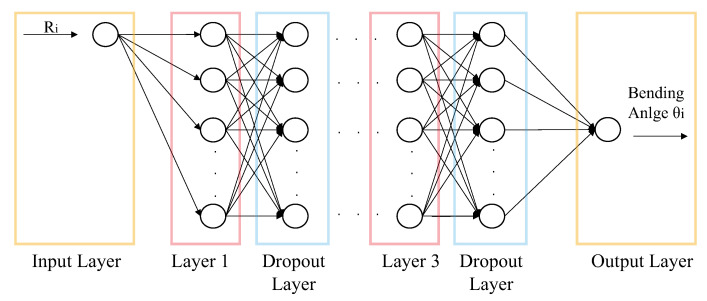
Structure of the calibration neural network based on a long short-term memory (LSTM) neural network. One input layer, three hidden layers with dropout layers and one output layer are included.

**Figure 6 sensors-22-07705-f006:**
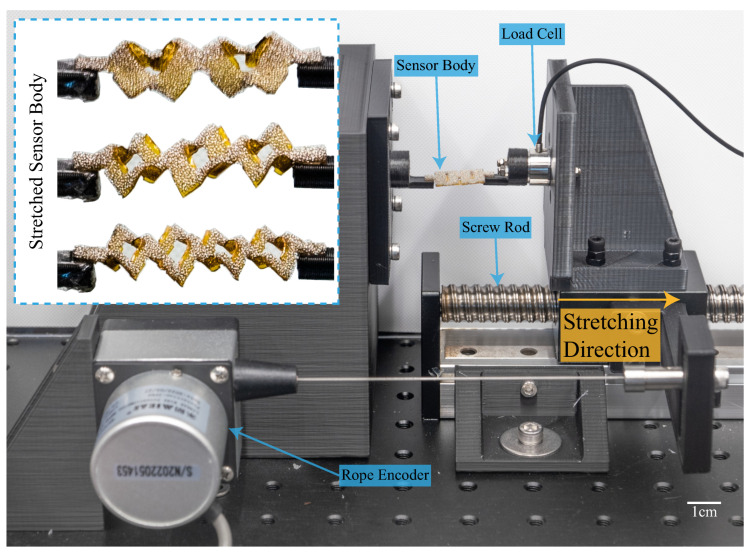
Experimental setup for sensor stretching experiment. Photos of a stretched sensor are shown in the top left corner. A scale bar with 1 cm is presented.

**Figure 7 sensors-22-07705-f007:**
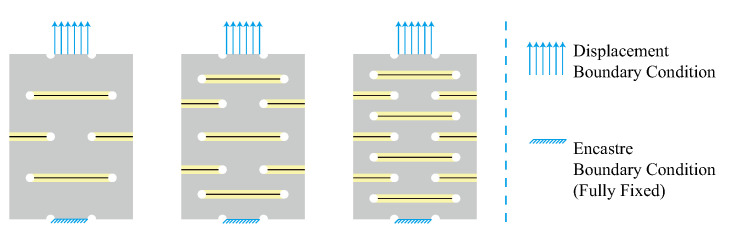
Schematic diagram of the boundary conditions setup in finite element analysis (FEA).

**Figure 8 sensors-22-07705-f008:**
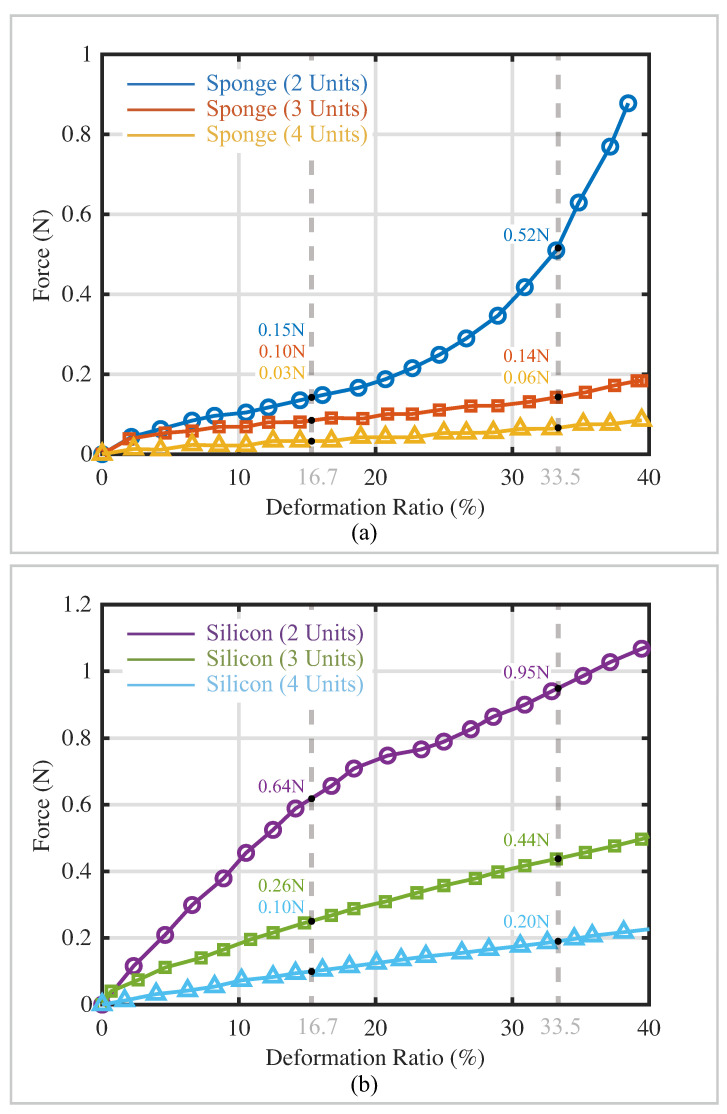
Results of the sensor stretching experiment. (**a**) Experimental results of the conductive sponge material. (**b**) Experimental results of the conductive silicon material. Resistive force of the sensors at a 16.7% deformation ratio (respective to a 45${}^{\circ }$ actuator overall bending angle) and 33.5% deformation ratio (respective to 90${}^{\circ }$ actuator overall bending angle) are indicated.

**Figure 9 sensors-22-07705-f009:**
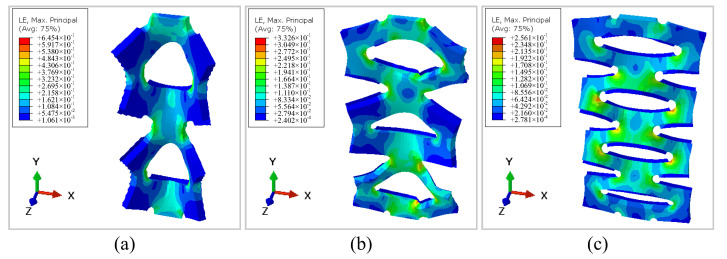
Finite element analysis (FEA) results of kirigami-inspired flexible sponge sensors with (**a**) a two-unit kirigami cutting pattern, (**b**) a three-unit kirigami cutting pattern and (**c**) a four-unit kirigami cutting pattern.

**Figure 10 sensors-22-07705-f010:**
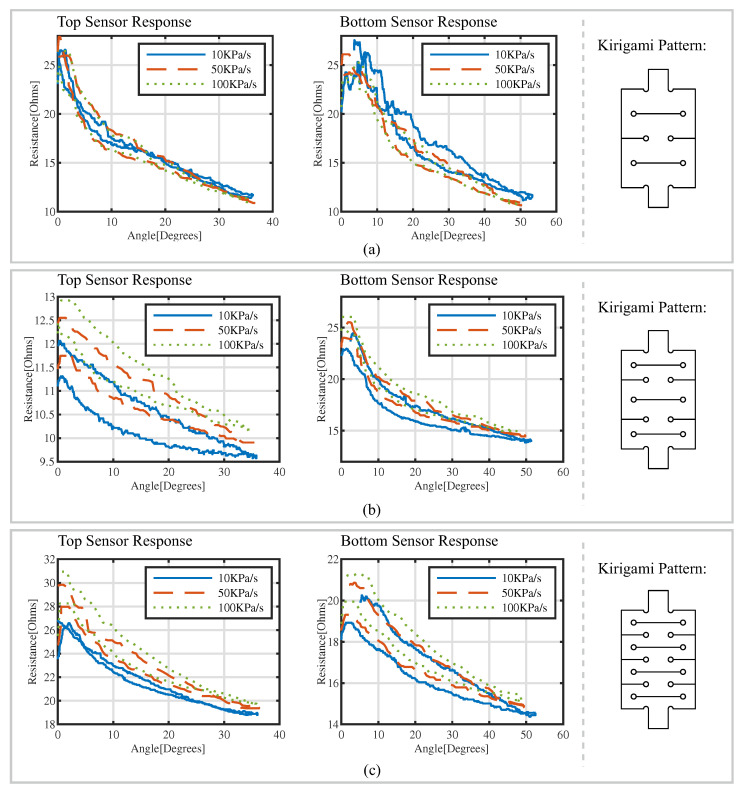
Resistance responses of sensors with (**a**) two-unit kirigami cutting patterns, (**b**) three-unit kirigami cutting patterns and (**c**) four-unit kirigami cutting patterns with respect to bending angles; 10 kPa/s (solid lines), 50 kPa/s (dashed lines) and 100 kPa/s (dotted lines).

**Figure 11 sensors-22-07705-f011:**
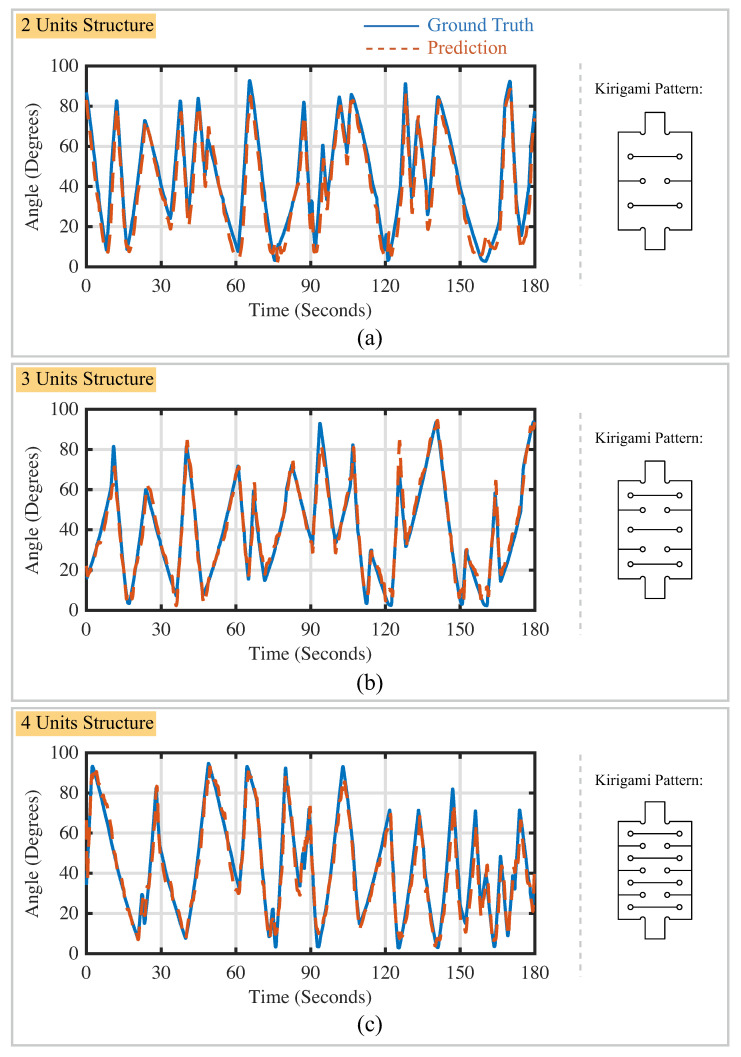
Performance of the LSTM calibration neural network in predicting the random motion of soft actuators. (**a**) Results for sensors with two-unit structure; (**b**) results for sensors with three-unit structure; (**c**) results for sensors with four-unit structure. Ground truth is indicated using a solid line. Prediction results are indicated using a dashed line.

**Figure 12 sensors-22-07705-f012:**
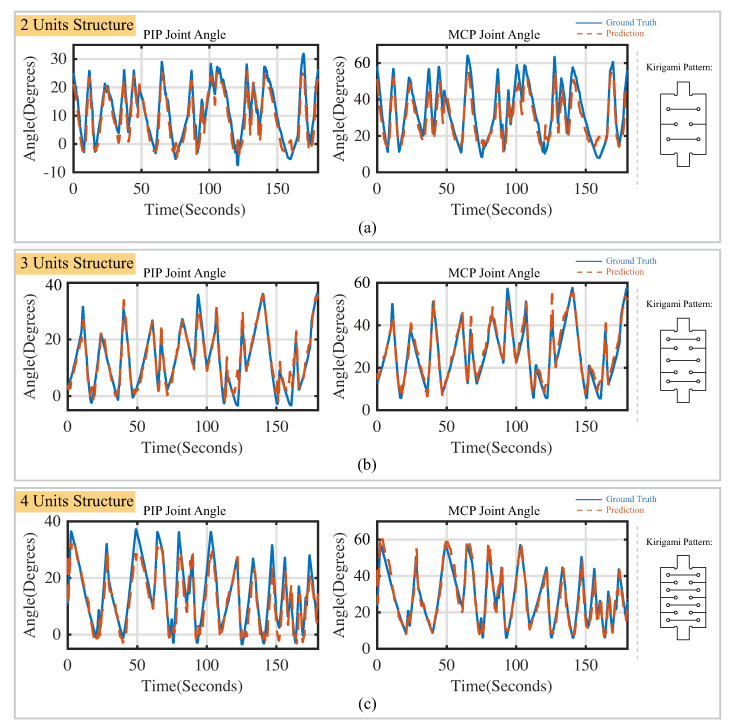
Performance of the LSTM calibration neural network in predicting joints angles of actuators in random motion. (**a**) Results for sensors with two-unit structure; (**b**) results for sensors with three-unit structure; (**c**) results for sensors with four-unit structure. Ground truth is indicated using a solid line. Prediction results are indicated using a dashed line.

**Figure 13 sensors-22-07705-f013:**
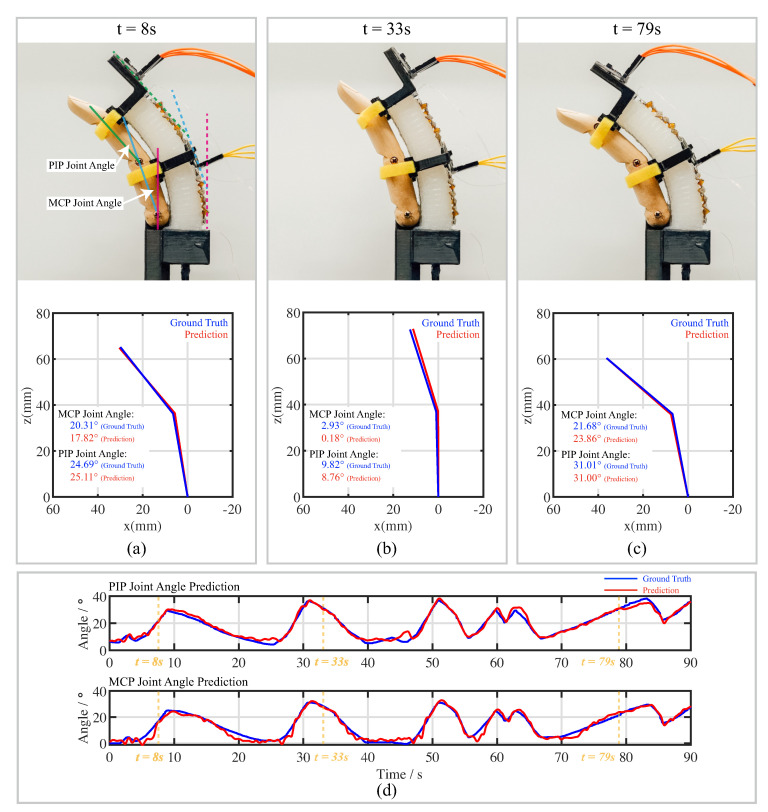
Finger-joint angle predictions using kirigami-inspired flexible sponge sensors. **Top**: photos of a wooden finger model with time step. Bottom: prediction of joints angles with ground truth. (**a**) Captured at t=8 s; (**b**) Captured at t=33 s; (**c**) Captured at t=79 s. (**d**) Prediction of finger-joint angles within a 90 s time-fram (time steps of finger postures in (**a**–**c**) are listed). Red lines: prediction. Blue lines: ground truth.

**Table 1 sensors-22-07705-t001:** Material parameters of PU in [41].

C10/MPa	C20/MPa	C30/MPa
3.76949	−0.415479	0.0289463

**Table 2 sensors-22-07705-t002:** Performance of the calibration neural network (all-to-one mode).

	RMSE (Degree)	Coefficient of Determination ($R^{2}$)
Two Units Structure	7.21	0.91
Three Units Structure	3.85	0.97
Four Units Structure	5.20	0.95

**Table 3 sensors-22-07705-t003:** Performance of calibration neural network (one-to-one mode).

MCP Joint Angle	RMSE (Degree)	Coefficient of Determination ($R^{2}$)
Two Units Structure	5.89	0.76
Three Units Structure	2.73	0.95
Four Units Structure	3.29	0.95
**PIP Joint Angle**	**RMSE (Degree)**	**Coefficient of Determination ($R^{2}$)**
Two Units Structure	3.21	0.87
Three Units Structure	2.51	0.93
Four Units Structure	3.85	0.85

**Table 4 sensors-22-07705-t004:** Resistive torque generated by sensor deformation.

Sensing Materials	Kirigami Patterns	Overall Bending Angle/Degree	Resistive Torque (mNm)	Resistive Torque Reduction (%)
Conductive Sponge	Two Units Structure	45	2.40	76.56
		90	8.32	45.26
	Three Units Structure	45	1.60	61.54
		90	2.24	68.18
	Four Units Structure	45	0.48	70.00
		90	0.96	70.00
Conductive Silicon	Two Units Structure	45	10.24	-
		90	15.20	-
	Three Units Structure	45	4.16	-
		90	7.04	-
	Four Units Structure	45	1.60	-
		90	3.20	-

## Data Availability

Not applicable.

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
