# Peer review of "Soft Robots’ Dynamic Posture Perception Using Kirigami-Inspired Flexible Sensors with Porous Structures and Long Short-Term Memory (LSTM) Neural Networks"

_sensors, 2022, doi:10.3390/s22207705_

Round 1

Reviewer 1 Report

This manuscript reports an end-to-end posture perception method using Kirigami-inspired flexible sensors and long short-term memory neural networks. The manuscript is well organized and written. However, this manuscript requires the following modifications:
1.-Abstract should consider the main results.
2.-This manuscript should include the main advantages and limitations of the proposed sensor compared to others reported in the technical literature.
3.-The authors should add more detailed information on the design of the proposed sensor and the test actuator.
4.-The FEM models should consider more information on the boundary conditions, load type, mesh type, and analysis conditions.
5.-The discussion of the main results must be improved.
6.-Which are the main challenges of the proposed sensor?
7.-Which are the future research works?

Reviewer 2 Report

This paper reports a piezoresistive sensing method to achieve posture perception of soft actuators based on a kirigami design. Posture estimation is an important problem in soft robotics. This paper tries to identify a solution to this problem, making little influence on the actuator motion. The kirigami design could offer large strain to the sensor. Authors verified the posture recognition capabilities in a bending actuator by utilizing long short-term memory neural networks. Overall, this paper is well-drafted with clear logic. The details on design, fabrication, and experiments are revealed in the paper. My suggestions to the paper are given as follows before publication.

1. The authors should clarify the contributions and novelty comparing to the previous work presented in a conference [23]. It seems that they use similar method.

2. The modeling of sensor deformation ratio is implicit. It would be better if authors could give a definition on it.

3. The figures seem not to be properly arranged. For example, authors mention Figure 1 on Page 8, while Figure 1 is arranged on Page 1. The authors should check them.

4. It would be better if authors use different symbols to identify different design, not only differentiating them in color. Same comments are given to Figure 9 and 10.

Round 2

Reviewer 1 Report

This revised manuscript was improved considering the reviewer's comments.